# Safe Food Handling Knowledge and Practices of Street Food Vendors in Polokwane Central Business District

**DOI:** 10.3390/foods9111560

**Published:** 2020-10-28

**Authors:** Khomotso J. Marutha, Paul K. Chelule

**Affiliations:** Department of Public Health, School of Health Care Sciences, Sefako Makgatho Health Sciences University, 0001 Pretoria, South Africa; majulisana86@gmail.com

**Keywords:** hygiene, street food vendors, knowledge, practices, South Africa

## Abstract

Street food vending is a common business practice in most South African cities. However, street vended foods may be a source of foodborne illnesses if their handling is not well regulated and executed. This study aimed to investigate the knowledge and practices regarding food safety by street food vendors in the Polokwane central business district. This was a quantitative descriptive study where a structured questionnaire was used to collect self-reported data from street food vendors. A checklist was used to collect observed data from the vending site on vendor practices and status of the vending environment. A total of 312 vendors participated in the study, most being young females aged below 40 years (62%), single (51.2%) with less than six years of trade experience (58.3%). Although the level of knowledge was high, safe food handling practices were mostly inadequate. Most vendors operated their businesses in the open air and tents (66.2%). Vending experience significantly correlated with safe food handling practices (*p* < 0.05). It is significantly riskier to sell cooked rather than uncooked food in the street. Lack of resources like water and a healthy environment negatively affected food handling practices. Health promotion on food safety is recommended for street vendors.

## 1. Introduction

Street food vending is a thriving business, especially in the developing countries, and it is one of the most common business practices globally, as it generates income in many of the low-income households [1]. Street food vendors are estimated to feed more than 50% of the urban population in developing countries [2]. However, street vended foods may be a source of many foodborne pathogens and illnesses if not regulated or properly handled. For example, in South Africa, there is no uniformity on the regulations for informal trading, which includes street food vending [3]. Knowledge of safe food handling and hygiene is vital for street food vending as it may reduce foodborne infections [4]. However, knowledge alone may not always lead to desired food hygiene practices, as other factors such as water resources, socio-demographics and cultural practices play a role as well [5,6].

Observed food handling practices in Nigeria show that nearly 47.6% of the street food vendors have poor hygiene practices as they serve food with bare hands, for example [7]. This practice has been similarly reported in Ghana, where almost two-thirds of street food vendors used their bare hands to serve food [8]. Most of the street vendors also have poor sanitary practices as they do not have immediate access to water. For example, 72% of street food vendors bought water from water vendors and the remaining minority (28%) obtained water from available taps nearby [9]. Thus, water challenges could result in street food vendors recycling water for either hand washing or dish washing [7]. Notably, the context of safe food handling practices is shaped by a number of factors that include vending experiences, personal hygiene, cultural characteristics, availability of resources and the vending environment. For example, lack of running water, poor sanitary conditions, as well as improper food handing have been associated with food borne illnesses such as diarrhoea, vomiting, abdominal cramps and nausea [10].

Regulation of this industry is not well established within South African municipalities, although street food vending needs to be monitored and controlled for the purpose of health and safety of the consumers. Thus, this study was conducted to investigate the level of knowledge, self-reported and observed practices regarding food safety, and to describe the environmental context that affects food handling practices of street food vendors in an urban South African municipality. The findings from this study may immensely contribute to the design of health-promotion programmes, targeting safe food handling practices in street food vending.

## 2. Materials and Methods

### 2.1. Design of the Study

This was a quantitative descriptive study, where a structured questionnaire was used to collect data from street food vendors in the Polokwane central business district. The structured questionnaire and the data extraction checklist, used to collect data, were adapted from similar studies done previously, reporting on the level of knowledge on food safety and safe food handling practices of street food vendors [1,7,8,11,12,13]. The English questionnaire tool was translated into Sepedi, a language spoken by the street food vendors in the Polokwane central business district. The Sepedi questionnaire was administered to the study participants orally by the researcher. The questionnaire in the current study was designed to collect data on socio-demographics, the level of knowledge on food safety and self-reported handling practices. The tool was pre-tested for validity by using it to collect and analyse data from 20 street food vendors. Analysed data from tool pre-testing was not included in the study sample data. A walkthrough into the vending site was also carried out to assess the hygienic status of the vending environment, availability of resources and the infrastructure, and to observe the safe food handling practices of street food vendors. Data from the vending site walkthrough was collected using a structured data extraction checklist. The data extraction checklist consisted of statements that described the status of the vending environment and the researcher’s own observation of vendors’ food handling practices.

### 2.2. Sampling

Polokwane Municipality had approximately 2500 street food vendors in 2006 [14]. Assuming that the street vendors had grown with the same increases as the total population, estimated by Statistics South Africa in the years 2006 and 2016 at the rate of 2.13%, over the same years, the street food vendors should have grown by 21.3%, corresponding to 3032 units.

The sample size of the population was calculated using the Raosoft sample size calculator, using the 95% confidence level, 5% error of margin and 50% response distribution, from the estimated population of 3032 street food vendors. Thus, a sample size of 342 study participants was obtained. Data collection was conducted from 17 July to 31 August 2017. Systematic random sampling was applied to select every consenting second participant repeatedly until the required sample size was achieved.

### 2.3. Ethical Issues

Ethical clearance to conduct the study was issued by Sefako Makgatho University research and Ethics Committee (SMUREC). Informed consent was also obtained from the street food vendors who were willing to participate in the study prior to data collection.

### 2.4. Data Analysis

Stata statistical software for data science (version, StataCorp LLC, College Station, TX, USA) version 16 was used to analyse data. Descriptive statistics was used to summarise data in the form of frequencies and percentages and presented in tables and figures. The level of knowledge and food safety practices were evaluated by the frequency of correctly answered questions. Pearson’s chi-squared test was used to test the association between cooked foods, observed food handling practices and the status of the vending environment. The outcomes with p-values less than 0.05 were considered statistically significant.

## 3. Results

### 3.1. Socio-Economic Characteristics

In this study, a total of 312 valid questionnaires were obtained (a response rate of 91.2%). Most of the street vendors (59.7%) were female, and nearly 62% of them had achieved secondary school education. The majority of these vendors (61%) were less than 40 years old, ranging from 14 to 71 years, with a mean of 38.6 years. Over half (54.5%) of the street food vendors had fewer than two children. The majority of them had a vending experience ranging from a couple of months to five years (*n =* 182, 58.3%) and only a small proportion (7.4%, *n =* 23) had a vending experience spanning over 20 years. Of the 312 interviewed street food vendors, 64.4% (*n =* 201) were registered with the municipality while the rest were not (Table 1).

### 3.2. Knowledge of Food Safety and Training

Street food venders were asked a number of questions to assess their knowledge of food safety. The majority of food vendors acquired their knowledge on food preparation through observation (*n =* 274, 89%) and the rest through formal training. Additionally, 75.4% (*n =* 233) of the street food vendors believed that formal training on food safety was necessary for their business purposes, while 16.2% responded otherwise. When safe food handling questions were asked, 89.6% (*n =* 277) of the vendors knew that washing hands before handling food reduced food contamination and almost all participants (99.3%, *n =* 303) correctly knew the importance of checking the expiry date of ingredients on the container labels before using them. The venders also correctly knew that re-selling and use of leftover food from the previous day was not a safe practice (Table 2).

### 3.3. Self-Reported Food Handling Practices by Street Food Vendors

In order to assess their safe food handling practices, the vendors were asked a number of questions, such as where they obtained water and whether they washed hands before handling food, for example. Reportedly, 35.6% (*n =* 107) of the vendors bought water for use, 24.6% (*n =* 74) accessed water from nearby taps, 19.9% (*n =* 60) from toilet taps, and 14.0% (*n =* 42) from other facilities which included nearby schools, shops, health departments and police stations. However, the minority (*n =* 18, 6.0%) of the vendors received water from municipal water tanks.

Up to 88.0% (*n =* 264) of the street food vendors reportedly washed their hands before handling food, while the rest never did. Although 82.1% of the street food vendors also reportedly washed raw food correctly before cooking and selling, 45.2% of them used the same water used for the raw food to wash other washable ingredients. Only 63.5% of the vendors washed their utensils after using them and 59.3% reported that they used the same utensils to serve different foods (Table 3).

When food vendors were asked what they did with left-over food, they reportedly stored it in a refrigerator or in a store-room with the intention of re-selling it to customers the following day, took it home for consumption by their family, donated it to other people or disposed of it as waste (Figure 1). Interestingly, the vendors re-sold food from the previous day although they knew that this was not a safe food practice.

### 3.4. Status of the Vending Environment and Food Handling Practices by Observation

In order to further understand the food handling practices in a vending environment, a checklist was used to observe the conditions of the vending site and how the vendors practically carried out their vending work. The items to be observed on the checklist included availability of basic resources, the types of food sold and the type of vending site structures (if there were any).

#### 3.4.1. Types of Food Sold by the Street Vendors

Through observation during the site walkthrough, slightly more than half of the vendors (*n =* 161, 51.6%) sold mainly cooked food which included pap (a South African cooked maize meal), meat, fish, salads, cooked vegetables and fried chips. Fruits and vegetables were sold by 110 (35.3%) of vendors, while the rest (*n =* 41, 14.1%) sold readily packed food items such as sweets, dried snacks and cool drinks.

#### 3.4.2. Status of the Vending Sites and Environment

A walkthrough into the vending environment determined how clean the stalls were and the state of maintenance, availability of resources such as water and waste disposal bins, how vendors handled the food they sold and if they followed safe food handling practices. In terms of the vending sites, a substantial number of the street food vendors (*n =* 113, 36.2%) operated their businesses in the open air, 31.1% (*n =* 97) operated from tents, while the minority (*n =* 15, 4.15%) operated under other shelters such as umbrellas and gazebos (Figure 2). Regarding the cleanliness around the stalls, 55.8% (*n =* 169) maintained their vending stalls in a clean condition. However, the majority of the stalls (56.7%, *n =* 170) were not well protected from the sun, pests, or animals. Protection of the vending site depended on the years of experience in the food vending business. This difference was statistically significant (Table 4). Garbage disposal remained one of the major challenges as 11.8% of the street food vendors left their garbage on site and did not dispose of it in waste bins provided by the municipality, but a substantial number (38.0%) used paper boxes to dispose their waste generated during the course of their business. Garbage on the site can impact on food safety practices as flies can breed in it, and thus food can be easily contaminated.

#### 3.4.3. Observed Safe Food Handling Practices of Street Vendors

Based on the researcher’s observation of food handling practices, 76.1% (*n =* 217) of street vendors were handling food with their bare hands (a practice not recommended), while 85.1% (*n =* 183) stored raw, partially cooked and cooked items separately, which is a recommended practice. While observing the handling of utensils, less than half of the street food vendors cleaned their utensils adequately after every use with soapy water (*n =* 127, 48.5%). However, 70.8% (*n =* 216) did not have access to portable water on their vending sites and this could have contributed to less cleaning of utensils. Additionally, 84.2% of the street food vendors were observed to be handling food while subsequently handling money. A minority of the street food vendors (*n =* 32, 10.8%) smoked cigarettes while preparing food. The details of the observed practices are shown in Table 4.

It was also observed that vendors’ experience in the trade business played a great role in safe food handling practices. Thus, the association between experience (years of trade) and food handling practices was statistically tested. The statistical analysis outcome showed that trading experience had a statistically significant association with vendors’ site of trade being protected from the sun, animals, or pests. Also, the practice of selling cooked food was more risky to public health than uncooked food, as the former gets easily contaminated if the vending site was not protected from the sun, animals and pests (*p* = 0.001), inadequate handwashing facilities (*p* = 0.000), handling food with bare hands (*p* = 0.000) and inaccessibility to portable water at or close to the vending site (*p* = 0.000). Details of these observations are shown in Table 4.

## 4. Discussion

The socio-demographic data in this study shows that the majority (60%) of the respondents were females. This figure is slightly less than that reported in other studies, where the majority of street food vendors were females, with a higher proportion of 78% and 95% respectively, in comparison to their male counterparts [8,11,15,16]. The female predominance of the vending business could be due to the fact that females are the leading bread winners in the majority of the households. About half of the street food vendors were single, as most of them were still young and had just completed matric level of education, while some lacked adequate basic education qualifications to further their schooling. This observation supports findings in a study done elsewhere, which also reported that over 50% of the street food vendors in their study were single [17]. This may not always be the case, as one Nigerian study, for example, found that the majority of street food vendors (58%) were married [15].

In this current study, in terms of years of trade, the majority of the street food vendors were in the category of 0–5 years and this is consistent with other study findings in West Africa [8], reporting that the majority of street food vendors were those with less than five years of work experience. This range was, however, higher in Nigeria, where the majority of food vendors were in the 6–10-year experience category [15]. One could argue that the reason why the years of service were mostly within the five-year category, is because most of the street food vendors were young, single school-leavers below the age of 35 years. Furthermore, lack of jobs and social deprivation as well as poor socio-economic status could also contribute to the high rate of the youth being self-employed as street food vendors.

It is very important to know how street food vendors acquired their knowledge on food preparation in order to establish their knowledge on food handling [18]. In this study, the level of knowledge was high as most questions were answered correctly by over 60% of vendors. The majority of the street food vendors acquired their knowledge on food preparation and safety through observation. Literature, confirmed by the findings in this current study, reportedly shows that the majority of the street food vendors acquired their knowledge through observation [15], or by learning from their family members [19]. Additionally, the majority of the street food vendors in this study knew that food safety training was necessary for essential safe food handling practices. This could mean that a vast majority of street food vendors around the world recognise the need to undergo formal training on food safety, a venture that most vendors in this study could not afford. Currently, only a minority of these traders are able to complete a formal course on food safety [19]. Furthermore, it was encouraging to learn that almost all the street food vendors (over 70%) correctly knew that it was unsafe to re-sell food from the previous day, and that re-freezing defrosted food was also an undesirable food handling practice that rendered food unsafe for public consumption.

In this study, we collected data on food handling practices based on both self-reported responses and the researcher’s own observation. Each of the data collection approaches supplemented the other to give a complete picture of vendors’ food handling practices. Although a high level of knowledge was demonstrated by the majority of the street food vendors in the study, some of this knowledge did not necessarily translate well into safe food handling practices as some street food handlers smoked while handling food and others handled money and food at the same time. While lack of water for street food preparation remains one of the major concerns in street food vending, street food vendors did not have access to waste disposal facilities. Thus, it was a big challenge for street food vendors to clean their utensils with re-used water every time after being used, with the result that their safe food handling practices were compromised. Lack of access to portable water is an observable trend in many studies of a similar nature [18,20]. The majority of the street food vendors bought the water they used for cooking and washing from water vendors, while a minority obtained their water from nearby facilities such as schools, shops and departments such as the health clinics. Inaccessibility to water was not always a problem in all areas of the country as some studies reported easy access to tap water [16].

Notably, most South African vendors in this study re-sold left-over food to the public the following day. This practice was observable despite the knowledge of the vendors depicting that it was unsafe to re-sell left-over food. This practice is also common in other countries like Kenya, where the majority of the street food vendors re-sold food left-overs to customers the next day [9]. It is encouraging, however, to learn that some vendors took the left-over food home for household consumption or donated it to charity. Additionally, most vendors in this study demonstrated an unhealthy practice by not covering food in sealed containers to prevent exposure to dust and flies. This practice is not common only in South Africa, as most vendors in other countries such as Haiti reportedly did the same [20]. This is not helpful especially given that the majority of the street food vendors in this study operated their businesses in the open air. Poorly protected or uncovered stalls do not give proper protection to the street food from dust, pests and smoke from vehicles [18]. Thus, safety of cooked food, in particular, is compromised, particularly if it is not covered.

Regarding the observed food handling practices of the vendors, most of them covered their hair at all times and their nails were neat and short. This contrasts with reports from other studies that reported on vendors who never covered their hair all the time [7,11]. In this study, we also observed that some of the street food vendors handled food with their bare hands. This poor food handling practice, for example, could result in increased chances of cooked food being contaminated by pathogens that are invisible to the eye. Contaminated food could lead to consumer acquisition of foodborne illnesses such as diarrhoea, nausea and vomiting [7,18]. Thus, there is an urgent need to sensitize the vendors on the importance of maintaining acceptable food hygiene standards [11].

The types of street foods sold are dependent on the geographical location of the street food vending. The majority of the foods sold in the study were cooked foods, which included pap, meat, fish, salads, cooked vegetables and fried chips. Most of these foods are cooked or prepared on the vending site. This study findings support in published reports elsewhere in South Africa, where similar South African dishes were sold involving, at most, cooked, ready-to-eat dishes [12,21]. In this regard, street vended foods differ from country to country given the variety of traditional dishes available in each country [18]. Cooked, ready-to-eat food requires specialised food handling practices and clean, enclosed environments, which were lacking in this study. This is because they are safer in a closed environment free from dust, sun heat and microbes. Also, there should be access to water for washing hands and utensils. These requirements were inadequate in our study setting.

## 5. Limitations of the Study

This study was conducted in an urban setting (central business district). Findings therefore may not be similar to other settings such as schools and construction sites, for example. The methodology of collecting self-reported data on vendors’ food handling practices could have been better accomplished using a Likert scale, rather than “Yes” or “No” responses. The study data was collected from one city (Polokwane) with limited resources, such as water. These findings may not be the same for another urban setting where basic resources, such as water, are more available. Thus, the study findings need to be interpreted with caution.

## 6. Conclusions

The study found that street food vendors have adequate knowledge on safe food handling practices, although knowledge does not always translate into practices. It is significantly riskier to practically sell cooked than uncooked food in the street. This risk is increased in part through vendor behaviour, while the unsafe vending sites contributed to the poor safety practices of street food vendors themselves. Restricted access to water impacted negatively on the daily operations of the business. Selling food in open space exposes food to microbes, heat and dust, which compromises food safety. However, the number of years in business was significantly associated with increased food safety practices. Public health promotion and training on food safety are recommended for the street vendors in the study setting.

## Figures and Tables

**Figure 1 foods-09-01560-f001:**
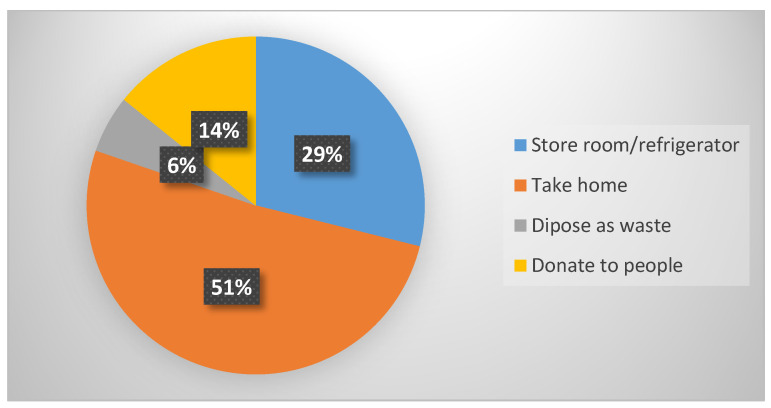
What food vendors did with their cooked left-over food.

**Figure 2 foods-09-01560-f002:**
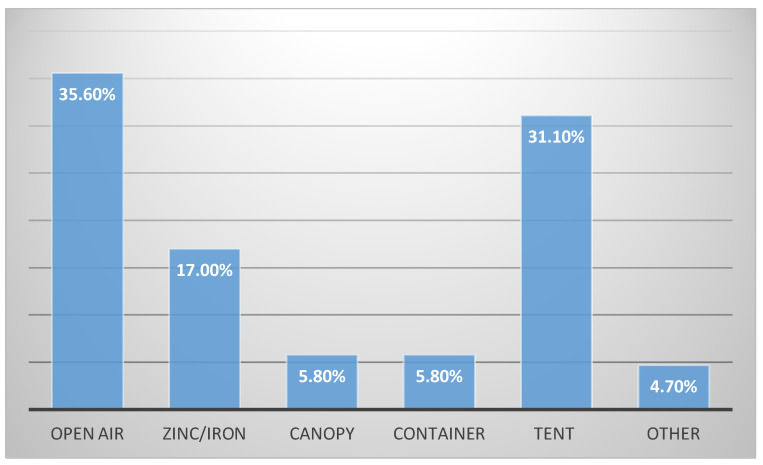
The structural status of the vending sites.

**Table 1 foods-09-01560-t001:** Demographic characteristics of the street food vendors (*n =* 312).

Variable	Level	Frequency	Percentage
Sex	Female Male	186 126	59.6 40.4
Age	<20 21–30 31–40 >40	8 96 86 122	2.6 30.8 27.6 39.0
Marital status	Single Married Widowed Divorced	155 93 40 15	51.2 30.7 13.2 5.0
Number of children	<2 3–4 >5	166 104 40	54.5 33.5 12.0
Number of dependants	<2 3–4 >5	190 100 22	60.9 32.1 7.0
Religion	Christian Non-Christian	210 98	68.2 31.2
Education	Primary school Secondary school Tertiary Never went to school	59 189 33 24	19.3 62.0 10.8 7.9
Nationality	South African Non-South African	273 33	89.2 10.8
Registration with municipality	Yes No	201 111	64.4 35.6
Payment of municipal fees	Yes No	202 107	65.4 34.6
Years of trade	0–5 6–10 11–20 >20	182 69 38 23	58.3 22.1 12.2 7.4

**Table 2 foods-09-01560-t002:** Street food vendors’ knowledge of food safety.

Knowledge Questions	Yes (%)	No (%)
I learned about food preparation through observation	274 (89)	33 (11)
I learned about food training though formal training	33 (11)	274 (89)
Is food safety training necessary for street vendors?	233 (75.4)	76 (24.6)
Do you know at least two municipal by-laws?	200 (64.3)	111 (35.7)
Are food laws helpful in regulating food safety?	186 (60.8)	120 (39.2)
Street food vendors should go for regular medical check-ups	236 (76.9)	71 (23.1)
It is important to check expiry date labels	304 (99.3)	2 (0.7)
Bloody diarrhoea can be acquired through contaminated food	275 (88.7)	35 (11.3)
Microbes can be found in the skin, nose and mouth of healthy food handlers	248 (80.3)	61 (19.7)
Typhoid can be transmitted by contaminated food	252 (82.1)	55 (17.9)
AIDS* can be transmitted by food	46 (14.9)	263 (85.1)
Washing hands before work reduces food contamination	279 (89.7)	32 (10.3)
Hand washing is necessary after touching money	261 (84.5)	48 (15.5)
Street food vendors cannot handle food while handling money	129 (43)	171 (57)
Reheating ready-to-eat food from the previous day for resale can contribute to food contamination	172 (69.4)	76 (30.6)
Re-selling and use of leftover food from the previous day is safe	40 (16.2)	207 (83.8)
I would stay at home when I have an infectious disease	233 (84.1)	44 (15.9)
It is important to change water used to wash utensils regularly	241 (94.5)	14 (5.5)
Covering food at all times improves the safety of food to customers	236 (92.2)	20 (7.8)
It is not a must to provide caps and gloves when you prepare food	246 (89.4)	29 (10.6)
Defrosted food can be re-frozen again	148 (60.9)	95 (39.1)

AIDS* = Acquired Immune Deficiency Syndrome.

**Table 3 foods-09-01560-t003:** Self-reported street food vendors’ food safety practices.

Safety Practices Questions	Yes (%)	No (%)
I always wash hands before handling food	264 (88)	36 (12)
I always wash hands with soap and water	110 (40.7)	160 (59.3)
I always wash hands with water only	106 (39.3)	164 (60.7)
I always wipe hands with wet cloth/wipes	54 (20.0)	216 (80.0)
I always wash utensils after serving food	155 (63.5)	89 (36.5)
I always wash raw food before cooking	220 (82.1)	48 (15.4)
I use the same utensils to serve different foods	150 (59.3)	103 (33)
I always wash utensils after serving food	155 (63.5)	89 (36.5)
I always use the same water to wash food ingredients	106 (45.2)	128 (54.8)

**Table 4 foods-09-01560-t004:** Association between cooked food, vending environment and observed food handling practices.

Checklist Variables	Checklist Observation		Statistical Evaluation	
	Yes *n* (%)	No *n* (%)	Pearson X^2^	*p*-Value
Vending site protected from sun, animals, or pests	130 (43.3)	170 (56.7)	11.77	0.001
Vending stall maintained in clean conditions	239 (83.6)	47 (16.4)	0.0412	0.838
Garbage bins on site	236 (80.3)	58 (19.7)	1.229	0.268
Toilet facilities on site	200 (64.7)	109 (35.3)	0.138	0.710
Environment around stall clean	167 (55.7)	133 (44.3)	1.440	0.230
Adequate hand washing facilities	81 (26.3)	227 (73.7)	26.333	0.000
Access to portable water at or close to site	89 (29.2)	216 (70.8)	31.267	0.000
Apron worn at all times	145 (47.1)	163 (52.9)	7.698	0.006
Food handled with bare hands	217 (76.1)	68 (23.9)	36.976	0.000
Hair covered at all times	197 (65.0)	106 (35.0)	1.440	0.230
Nails neat and short	252 (82.9)	52 (17.1)	0.851	0.356
Jewellery worn	49 (16.0)	258 (84.0)	1.276	0.259
Vendor uses same utensils to serve/prepare food	165 (68.2)	77 (31.8)	549.8	0.000
Raw and cooked food items are stored separately	183 (85.1)	32 (14.9)	7.775	0.005
Food stored or displayed in sealed containers	109 (42.2)	149 (57.8)	46.567	0.000
Utensils covered	88 (33.2)	177 (66.8)	46.4	0.000
Utensils cleaned adequately every time after usage	104 (41.8)	145 (58.20	49.2	0000.
Utensils cleaned with soap water	127 (48.5)	135 (51.1	59.8	0.000
Vendor handles food while handling money	251 (84.2)	47 (15.8)	19.018	0.000
Vendor smokes while handling food	32 (10.8)	265 (89.2)	1.810	0.178

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
