# Peer review of "Safe Food Handling Knowledge and Practices of Street Food Vendors in Polokwane Central Business District"

_foods, 2020, doi:10.3390/foods9111560_

Round 1

Reviewer 1 Report

The work investigates the level of hygiene demonstrated by street food vendors in Polokwane. The experimental design and methodology are appropriate, however the factors analyzed, or in any case stated, as trading experience, knowledge, availability of water and facilities, promotion actions, rules and control, socio-economic and cultural characteristics, characteristics of the surrounding area, type of food sold, preparations conducted at the place of sale, deserve to be better identified and contextualized, in order to highlight their role in determining the food handling practices. The same food handling practices have not been analyzed with regard to their real impact on the risk of food diseases, through the association with the type of preparation and the type of food sold.

The proposed analysis supplement would have the ability to greatly raise the scientific value and innovation of the work.

Regardless of the parts reported as to be corrected in the text, I ask the authors to improve the description and explanation of the results, schematizing them in a more orderly way.

Parts of text to be corrected

10-11     “if not properly handled and regulated”. The articulation of the two concepts (handling and regulating) does not appear correct. I propose to modify in “if their handling is not well regulated and executed”.

16          “below” repeated;

18          “Safe”: correct in “safe”;

17-22     There are repetitions and the text seems not well articulated. I propose to modify as follows: “Although the level of knowledge was high, safe food handling practices were mostly inadequate. Trading experience significantly co-related with safe food handling practices (p<0.05). Most vendors operated their business in open air and tents (66.2%) Street food vendors in the study setting have adequate knowledge on safe food handling and hygiene and the lack of resources like water and healthy environment negatively affected safe food handling practices. Health promotion and training on food safety are recommended for the vendors.”

31          See what is reported in reference to rows 10-11.

46-47     “Regulation of this industry is not well established within South African municipalities. Thus, although street food vending is a thriving industry it needs to be monitored for health and safety of  the consumers”. The relationship between the lack of regulation and monitoring needs to be better articulated. I propose to modify in “Regulation of this industry is not well established within South African municipalities, although street food vending needs to be monitored and controlled for health and safety of  the consumers”.

55          “The structured questionnaire used to collect the data and was adapted …” Modify in “The structured questionnaire used to collect the data and was adapted …”

61-67     “Polokwane Municipality had approximately 2500 street food vendors in 2006 [14] Assuming that the annual growth rate of the total population has been constant, over the 10 year-period, based on statistics South Africa’s estimated annual population growth rate of 2.13%, the street food vendors too, would have increased by 21.3%. Thus, with 2500 of street food vendors in 2006, estimated population of street food vendors in the Polokwane municipality in 2016, with the assumption that the growth rate is 21.3% would be 3032.” The concepts expressed are articulated in a way that makes them unclear, with unnecessary repetitions. I propose to modify as follows: “Polokwane Municipality had approximately 2,500 street food vendors in 2006 [14] Assuming that the street food vendors have grown with the same increases as the total population, estimated by statistics South Africa's in the years between 2006 and 2016 at a rate of 2.13%, over the same years the street food vendors should have grown by the 21.3%, corresponding to 3,032 units.”

92          “.. had a the trading experience spanning over” Modify in “.. had a the trading experience spanning over”.

171-172 The explanation reports the results of other works, but while the works cited are four, the data only two! Could the attribution of bibliographic references be clearer?

181        “and this consistent with” The verb is missing.

186        “young, single school-leavers below 35 years of age” Punctuation should be correct.

193        “Literature, supported by the findings in this current study,” The term "supported" does not seem appropriate. “Confirmed”?

207        “it becomes, a”. Punctuation should be correct.

209        “after use, and thus the hygiene status is compromised.” It should be better “after use, With the consequence that the hygiene status is often compromised.”

224        “This practice in common in other countries like Kenya” Correct “This practice is common in other countries like Kenya”.

254        “This study setting is more in the urban central business district” the concept should be described more clearly.

Author Response

Comments from fisrt author were helpful and have helped to re-shape the manuscript for the better. 

  1. The factors presented and analysed in this study have been contextualized to be in line with the study objectives  Furthermore, food handling practices have been analyzed with their regard to the real impact on the risk of food diseases. This was addressed by re-analysing data based on the types of food  (cooked and uncooked) vended against other relevant variables. Thus the scientific value of the paper i has been enhanced.
  2. The parts of text pointed for corrections, line by line have been addressed by accepting the review suggestions or making changes as per the reviwer suggestions.
  3. Calculation of the sample size under methodology section has been ammended by accepting the reviewer suggestions.
  4. The manuscript was send in for English editing and some english related comments have been addressed.

Reviewer 2 Report

The topic of street food is of great interest, not only to Africa, but to Asia as well as Latin America and some areas of Europe. Besides hygiene, the cooking practices are also of importance. 

This paper addresses the issue of hygiene, as it seems to be of primary importance. My suggestion is to consider and furder  expand the study ( or perform another one) looking at the quality of meals sold.

Author Response

The comments by the reviewer have been noted and will  consider expanding the study to include the quality of meals.

I thank the reviewer for taking time to review the manuscript.

Reviewer 3 Report

The topic is very interesting and important. 

Title: Very clear

Purpose: clear

Material and methods: The observational survey is not clearly described in this part. You can read about it below Results. It should be shifted over to the Material and Method part.

According to the questionnaire it gives the scientific soundness a low score because the questions are answered by "yes" or "No".

It should be highlighted that some of the questions are self-reported. Exampel: "I wash hands before handling food"  is a typical question where you do not have any control. According to the publication "Redmond & Griffith (2003) a comparison and evaluation of reserach methods used in consumer food safety studies. International Journal of consumer studies, 27, 17-33" there are a large difference between selfreported data regarding behavior than observed data. This should have been discussed more in depth. A large part of the questionnaire are about attitudes and for these kind of questions there should have been used a Likert scale and not the answers "Yes" or "No". The choice of method make the entire study crucial.

The observational study must undoubtly be better described in the Material and method part!

Table 4 is unclear. What is "Level of adherence"?

Discussion. There are several weak points in the collecting of data i.e. self-reported data regarding behavor and not using Likert scale when it comes to questions regarding attitudes. These weak points should be better discussed more in depth.

Conclusion is based on the methods above. I would like to ask the authors regarding the conclusion: What new did you learn and what facts presented in the conclusion did you know before the investigation? It was indeed interesting the result regarding the number of years and business and increased food safety practices, however, could it be better presented in the manuscript and in the tables?

The reference list is fine, however, please check if the reference nr 10 i missing in the text?

Author Response

  1. The observational survey which was not clear has been addressed by re-writing the methodology section to reflect the same.
  2. The weakness of collecting data using "yes"  and "No" instead of the Likert scale has been noted and given as one of the study limitations. The discussion section has also given an explanation on the role of self-reported practices versus the observed handling practices. Both complement each other.
  3. Table 4 has been modified for clarity purposes.
  4. The conclusion has been re-written to reflect new insghts emanating from this study.
  5. Refence no. 10 checked and is included in the text.

Round 2

Reviewer 1 Report

Requests for further information and corrections were positively received. The updates made to the text have made the analysis more in-depth and the exposition clearer.

Author Response

I want to thank the reviewer for the constructive, positive comments.